

# Ice ridge density signatures in high resolution SAR images

Mikko Lensu and Markku Similä

Finnish Meteorological Institute(FMI), Marine Research, Erik Palménin aukio 1, 00560 Helsinki, Finland

**Correspondence:** Mikko Lensu (mikko.lensu@fmi.fi)

**Abstract.** The statistics of ice ridging signatures was studied using a high (1.25 m) and a medium (20 m) resolution SAR image over the Baltic sea ice cover, acquired in 2016 and 2011, respectively. Ice surface profiles measured by a 2011 Baltic campaign was used as ground truth data for both. The images did not delineate well individual ridges as linear features. This

was assigned to the random, intermittent occurrence of ridge rubble block arrangements with bright SAR return. Instead, the ridging signature was approached in terms of the density of bright pixels and relations with the corresponding surface profile quantity, ice ridge density, were studied. In order to apply discrete statistics, these densities were quantified by counting bright pixel numbers (BPN) in pixel blocks of side length $L$, and by counting ridge sail numbers (RSN) in profile segments of length $L$. The scale $L$ is a variable parameter of the approach. The other variable parameter is the pixel intensity threshold defining bright pixels, equivalently bright pixel percentage (BPP), or the ridge sail height threshold used to select ridges from

surface profiles, respectively. As a sliding image operation the BPN count resulted in enhanced ridging signature and better applicability of SAR in ice information production. A distribution model for BPN statistics was derived by considering how the BPN values change in BPP changes. The model was found to apply over wide range of values for BPP and $L$. The same distribution model was found to apply to RSN statistics. This reduces the problem of correspondence between the two density concepts to connections between the parameters of the respective distribution models. The correspondence was studied for

the medium resolution image for which the 2011 surface data set has close temporal match. The comparison was done by estimating ridge rubble coverage in 1 km$^2$ squares from surface profile data and, on the other hand, assuming that the bright pixel density can be used as a proxy for ridge rubble coverage. Apart from a scaling factor, both were found to follow the presented distribution model.

**Keywords**: sea ice, ice ridges, ridging parameters, Synthetic Aperture Radar (SAR), X-band, TerraSAR-X, statistical analysis, simulation.

## 1 Introduction

The Baltic Sea is a semi-enclosed brackish sea water basin in northern Europe. Baltic drift ice has dynamic nature due to forcing by winds and currents, which results in an uneven broken ice field with distinct floes, leads and cracks, brash ice

barriers, rafted ice and ice ridges. The upper limit for thermodynamically grown ice in the drift ice zone is 70 cm or less during



most winters (Palosuo et al., 1982), while the keel depth of ice ridges is typically 5 to 15 m (Leppäranta and Hakala, 1992; Ronkainen et al., 2018).

Navigation in the Northern Baltic continues trough the ice season. During an average winter all Finnish ports become surrounded by ice and they are kept accessible by icebreakers during severe winters also. As there are annually up to 20,000 port
calls trough the ice cover in Finland only, there exists a demand for accurate ice information especially on ridging conditions. Without the presence of ridged ice the managing of wintertime maritime operations would be much simpler matter.

Ridge fields complicate the navigation and icebreaker assistance by inducing performance variations between ships of different capabilities as well as speed variation for each individual ship. This includes occasional besetting, which, in convoy operations, increases collision risks and causes delays and disorder in port logistics. Better information about the distribution
of ridged ice would improve routing, icebreaker operation planning and the predictability of arrival times. This would make the Baltic winter navigation system as a whole more efficient, safe, environmentally friendly and economical. This applies also to other to other ice infested sea areas.

Operative production of ice ridging information is based on satellite data and surface observations. In the Finnish-Swedish ice charts this information is coded by the degree of ice ridging (DIR), which is a numeral that seeks not only to quantify ridging
but also to characterise navigational difficulty. Icebreakers communicate their estimates of DIR values to the ice services where DIR values are assigned to ice chart polygons using the estimates, other data, and manually interpreted SAR images.

Due to the qualitative nature of DIR the need for truly quantitative ridging statistics persists. The usual surface parameters to describe ridging are ridge height and ridge density which is defined as the number of ridge sails per km along a linear track. The surface statistics can be linked to the statistics of the subsurface ridge keels with cross-sectional models for sail and keel
geometry. This provides from the surface parameters an estimate of the total mass of ridge rubble which is a key quantity in dynamic ice models. Statistical models can also be used to generate simulated thickness profiles for ridged ice, applied to model ship speed reduction and ship besetting risk (Kuuliala et al., 2016). Finally, ridge sail height and ridge density are related to the fraction of the surface area covered by ridge rubble, a parameter which contributes to the magnitude of $\sigma^o$ in SAR images.

Ridge sail height and density can be determined by airborne profiling measurements which however do not usually belong
to the routines of ice information production. At the same time, research seeking to determine ridging parameters from SAR images has not made enough progress to fulfill the expectations of the ice information producers. This difficulty has three main sources. Individual ridges are usually not resolved, the characteristics of the satellite instruments, images and acquisitions are varying, and there are sensitivities to ambient conditions and surface properties not related to ridges. To this adds the fact that ground truth data and the SAR signatures cannot usually be matched if the ice has drifted or deformed in the meantime.
In this paper we approach the problem in a way that makes feasible the quantification of ridge density from SAR at least in a relative fashion. The approach relies on the assumption that ridge density variation dominates the spatial variation of ridging signature in SAR images. The main results is that the same statistical model is found to apply both to the variation of ridge density and to the variation of density of pixels chosen to represent ridged ice. An extensive set of Baltic surface profile data is used to derive the statistical model for ridge density. Two different TerraSAR-X X-band SAR images from the Baltic Sea
are used, obtained with different imaging modes and resolutions. The high resolution (1.25 m) image is used to study how





individual ridges contribute to SAR signatures, and demonstrate that statistical model for ridge density applies also in SAR context. The other, with 20 m resolution, is nearly concurrent with the surface data set which is used to validate the approach.

## 2 Background

SAR-based retrieval of ridging has previously approached the problem from two main directions. Physics based approaches have tried to determine the microwave backscattering properties of ridged ice types so that these could be distinguished from other ice types. The scale of this problem ranges from the size of ridge blocks to the pixel size of low resolution SAR images for which the ridged ice $\sigma^\circ$ comes from a mixture of block accumulations and other ice types. On the other hand, image based approaches seek to develop methods to retrieve degrees of ridging using image segmentation methodologies, supported and validated by field data.

One reason for the weak connection between the physics based and image based approaches is that it is not well understood how the pixel-to-pixel intensity variations relate to the ridging statistics and how this is affected by different wavelengths, polarisations, viewing angles, platforms, resolutions, roughness types, and ambient conditions. Backscattering from rafted ice, brash, sastrugi and other rough surfaces may overwhelm the ridging signature, and changes in temperature, moisture and snow cover may alter the discernibility of ridging signatures from day to day. Theoretical models, e.g., (Albert, 2012), predict that for high frequency waves the volume backscattering may exceed surface backscattering and significantly change the $\sigma^\circ$. A feature specific to low-salinity ice is the importance of scattering dominated by volume inhomogeneities in the uppermost part of the ice. As verified by (Dierking, 1999) this partly explains the high variation in X-band $\sigma^\circ$ values for the Baltic Sea ice for which the salinity ranges from 0.2 to 2 permille (Hallikainen, 1992).

Not many papers have addressed the identification of ridges from SAR imagery. The likely reason is that the resolution of the operative SAR satellite modes has been too low, usually not better than 100 m, to resolve individual ridges. Instead, $\sigma^\circ$ depends on the aggregate effect of highly variable conditions in each pixel. For an experienced eye it may appear evident which texture types indicate ridging even in very low resolution imagery. The step from this to quantitative methods has proved hard although correlations clearly exist. An article which discussed detecting ridges in length is (Melling, 1998) where the study area was in the Beaufort Sea. A strong correlation was observed between the spatial frequency of ridges identified from the SAR and the average draft of the ice field in the X-band. It was noted that the X-band airborne SAR (25 m resolution) yielded better ridge identification than the C-band ERS-1 SAR (30 m resolution) for first-year pack ice.

In the Baltic Sea the most intensive phase of the SAR based work related to sea ice ridging took place in late 1980's and early 1990's. Several aspects of the problem were examined. There were both physics based approaches seeking to model the microwave backscattering from ridges, and image based approaches trying to identify ridging signatures from SAR images or classifying them with respect to degrees of deformation. Backscattering models were constructed by (Johansson et al., 1992; Manninen, 1992; Carlström and Ulander, 1995; Manninen, 1996; Carlström et al., 1997). These included also three-dimensional models taking full ridge geometry in account, that is, the shape of the ridge sails and the size, shape and angle distributions of the sail blocks. According to the 3D-backscattering model for an ice ridge by (Manninen, 1992) the most



important ice properties in the C-band microwave surface backscattering from ridge sails are, in order of importance, microscale surface roughness, dielectric constant and geometrical properties of the ice blocks. If the surface roughness and dielectric constant variations over the ridged and surrounding areas are small the magnitude of backscattering is dictated by the geometry of the ice block accumulations, basically the relative surface area covered by the accumulations and the orientations of the ice

blocks in them.

In (Carlström and Ulander, 1995) the authors used 2D-model where an ice ridge is assumed to be a collection of circular facets with superimposed surface roughness. Thicknesses and slopes of facets varied. The authors concluded that the specular reflections are dominant unlike the results reported by (Manninen, 1992). According to (Manninen, 1992, 1996) the main difference between ridged and level ice is that backscattering from ridge blocks has a broad range of incidence angles whereas

level ice has a very narrow range. Both models predict rather similar results for first-year ridges in C-band SAR imagery due to the broad distributions of ridge-block orientations and dimensions. Ridge backscattering has also been observed to be slightly sensitive to radar azimuth angle by (Johansson et al., 1992). We note that the modelling predictions based on *in situ* measurements agreed well with SAR data in the comparison reported by (Carlström et al., 1997).

Due to the complicated nature of the geometric models they can hardly be utilised in practice to interpret observed $\sigma°$.

Sufficient understanding of ridging statistics, required for the integrating the physical model for areally averaged backscattering values, has been lacking as well. However, the physical models help us understand which ice properties control the separation of ridges from the surrounding level ice and why ridges often remain undetected even in high resolution SAR images or fail to appear in them as continuous curvilinear features.

In our SAR based ridging studies we have often indicated the intensity of ridging using the degree of ice ridging (DIR)

categories in the absence of field data for large sea ice areas. The DIR classes are semi-heuristic ice charting numerals. Still, a clear correlation between the classes and deformed ice volume was found for the profile 2011 data set used in (Gegiuc et al., 2018). The earliest SAR based ridging estimation in the Baltic Sea was (Similä et al., 1992) where a regression estimation for ridge density was proposed with tail-to-mean ratio as a predictor. Tail-to-mean ratio is a function of the SAR pixel value distribution It yielded reasonable results for the test set consisting of the *in situ* airborne profiling data but due to lack of further

validation data the model could not be adapted to different ice conditions. Later, the DIR estimation problem in the Baltic Sea has been investigated in multiple papers. In (Mäkynen and Hallikainen, 2004) the $\sigma°$ distributions were computed for several ice deformation categories with different incidence angles utilising data collected during many Baltic Sea scatterometer field campaigns. The utilised deformation categories approximated the DIR classes and were determined using a video-based assessment. Only small differences were noticed between the X- and C-band results or using different polarisations with

the exception of HV-polarisation. The results agree with those obtained in (Eriksson et al., 2010). Part of the scatterometer data collected by Mäkynen was reanalysed using a hierarchical Bayesian model in (Similä et al., 2001) with improved DIR classification results.

In (Gegiuc et al., 2018) sea ice ridging was assessed using the DIR classes retrieved from ice charts covering the whole Gulf of Bothnia. The analysis had three stages: segmentation of SAR image, computing a feature vector to each segment and then

classifying the segments. Training data consisted of several ice charts. Different ice charts served as the validation data for the





classification maps. The results were encouraging and the method will be transferred into operational use. Thus it is possible to improve regional ridging information by combining advanced methods with usual operative information production by expert analysts. The actual ridging statistics required for effective ice routing and for ice forecast model assimilation is not captured by the DIR classes however.

A case study by (Similä et al., 2010) showed that in dry and cold ice conditions with thin snow cover it was possible to find correspondence between freeboard and C-band SAR data in the Baltic Sea assuming that the dominant thickness of the regional level ice is known. The used field data was collected during the CryoVex-2005 campaign. A nonlinear regression model with three control variables, $\sigma^\circ$, the dominant thickness of level ice, and a variable accounting the effect of the SAR incidence angle, was used. The predicted estimates by the model followed closely freeboard changes also in the ridged area.

## 3  Approach

Considering the task to quantify ridging from a SAR image it is assumed here for discussion purposes that open water areas have been detected (Karvonen et al., 2005). It may also be assumed that the ice cover signature has been segmented by an algorithm or by an ice charting expert into ice types among which ridged ice is one and such that the segmentation excludes other rough types of non-ridged surface with very bright return. The present approach concentrates on the remaining task of

quantifying ridging for the ridged ice.

In a birds-eye view ridging shows as block accumulations covering the surface. These are often identifiable curvilinear sails with approximately triangular cross section and with typical scale 1-10 m in across sail direction and tens to hundreds of meters in along sail direction. The essential feature of ridge rubble is that it may generate a strong backscattering signature. Thus all block accumulations with block structure resembling that of ridge sails are understood here as ridging. This includes

rubble fields and other more chaotic block formations that cannot be decomposed into curvilinear sails. From backscattering viewpoint the the principal ridging quantity is the coverage (relative area) of ice surface covered by ridge rubble.

It appears clear that the returns from the ridge rubble dominate the brighter end of the backscattering intensity histogram. The brightness statistics involves certain uncertainty, however. The intensity histogram depends on the processing of the image and on enhancements aiming at good visual appearance. The ambient conditions affect the brightness statistics so that temporally

separated SAR images from the same ridged ice field may have quite different intensity histograms. However, an ice charting expert could still recognise the same ridging features and would provide a similar manual classification to 'degrees of ridging' in both cases. The signatures that suggest the presence of ridges create patterns that appear as persistent and symptomatically nonhomogeneous. This shows up clearly in binary images that comprise a certain percentage of bright pixels from the intensity histogram tail. These often delineate the areas that the ice expert would select as ridged ice. If the percentage is reduced, the

patterns become more sparse but tend to retain structural congruence with the non-reduced patterns.

In a way, the objective of the present approach is to provide statistical foundation for the assessment that the ridging signatures in two SAR images 'look the same'. Towards this end, a certain bright pixel percentage (BPP) is selected with an intensity threshold. For higher BPP the selected bright pixels are expected to be predominantly returns from ridge rubble. The spatial





variation related to the ridging signatures is described in terms of bright pixel number (BPN) in pixel blocks with side length $L$. This is equivalent to bright pixel density BPN/$L^2$ in scale $L$ but the use of BPN is preferred as it is additive between blocks and the statistics is discrete. The same approach can be defined for ground truth profile data in terms of ridge sail numbers (RSN) in profile segments of length $L$, equivalently ridge density RSN/$L$ in scale $L$. The scale $L$ and the BPP are variable parameters

of the approach, as well as the cutoff threshold for ridge sail height that corresponds to the intensity threshold for the BPP.

The remaining paper is organised as follows. In Section 5, after presenting data and weather in Section 4, the ridge signature visibility and BPN concept is examined using the high resolution SAR image. First application oriented 'contextual images' obtained by a sliding BPN operation are presented. Observations on how the BPN values change in BPP changes are then analysed. This leads in Sections 6.1-6.3 to a statistical model that is derived by considering a process, called threshold process,

where the the change of BPN or RSN is related to the sequential decreasing of intensity threshold or sail height cutoff threshold by small steps, respectively. The model is then validated both for the ground truth dataset RSN in Section 6.4 and for the high resolution image BPN in Section 6.5. This in done by fitting the model to the observed statistics and, more fundamentally, by directly validating the generative threshold process assumption of the model. The result reduces the problem of linking BPN and RSN statistics, or equivalently bright pixel density and ridge density statistics, to the establishing parameter connections

between the two. In Section 7 the model is further validated and the parameter connections are established for the medium resolution image that is concurrent with the ground truth profile data set.

## 4   Data Sets and Processing

### 4.1   SAR data

The German TerraSAR-X (TSX) satellite was launched in 2007. It can be operated in different modes, usually in Stripmap

or ScanSAR mode (Fritz and Eineder, 2013). The ScanSAR HH-polarized image was acquired on 28 February 2011 over the northern part of the Sea of Bothnia near the Quark. It has swath width of 150 km and azimuthal length of 180 km. The ScanSAR mode consists of four Stripmap beams which are combined to achieve the 150 km wide swath. The preprocessing of the ScanSAR image comprised calibration (calculation of $\sigma_{HH}^\circ$), georectification and land masking. The image was rectified into the Mercator projection with 20 m pixel size. This georectification is compatible with the FIS ice charts and the navigation

system of the Finnish and Swedish icebreakers. In this Mercator projection the reference latitude is 61°40' N. The incidence angle varies from 29.5° to 38.7°. Because the incidence angle range is relatively narrow, we did not perform any statistical incidence angle correction (Mäkynen et al., 2002), only the calibration of the backscattering coefficients $\sigma^\circ$. The equivalent number of looks (ENL) was on average 7 in the ScanSAR scene and the radiometric resolution 1.4 dB. The average noise equivalent sigma zero (NESZ) was -21 dB (Fritz and Eineder, 2013).

In this study we extracted a 106 x 94 km subimage ($5300 \times 4700$ pixels) for the statistical analysis. The size of the selected subimage was dictated by the available nearly simultaneous laser profile data.

For high resolution SAR studies is used a Stripmap image acquired on 5 March 2016 near the Hailuoto island in the Bay of Bothnia . The acquired product is a Geocoded Ellipsoid Corrected (GEC) product without any terrain correction. It is also a





Spatially Enhanced Product (SE) designed for the highest possible square ground resolution. The HH-polarization image was
rectified into 1.25 m resolution in the Mercator projection. The covered area is about $33.6 \times 42 \text{km}^2$ (width $\times$ length). For the
purposes of the paper, a 19x19 km ($15200 \times 15200$ pixels) subimage of the Stripmap image was used. The locations of both
high and medium resolution images are indicated in Figure 1 on the ice charts for the days of acquisition.

## 4.2 Laser data

The laser data was collected during a helicopter-borne thickness profiling campaign in the Bay of Bothnia during $2 - 7$ March
2011, i.e., during the week after the acquisition of the ScanSAR image. The measurement system combines laser distance mea-
surement to ice or snow surface and electromagnetic (EM) distance measurement to the ice–water interface. The measurement
system was similar to that described by (Haas et al., 2009). The campaign comprises approximately 2800 km of helicopter-
borne EM (HEM) measurement lines shown in Figure 1. After leaving the flight base the logging was turned on at about
64 N 23 E and within 50 km from this point the repeating tracks sample the ice cover almost completely in 1 NM resolution.
Only the surface data is used here. The thickness data has been addressed in (Ronkainen et al., 2018), and the relationship of
the surface ridging parameters to the thickness and to the DIR ridging index of Finnish-Swedish ice charts in (Gegiuc et al.,
2018).

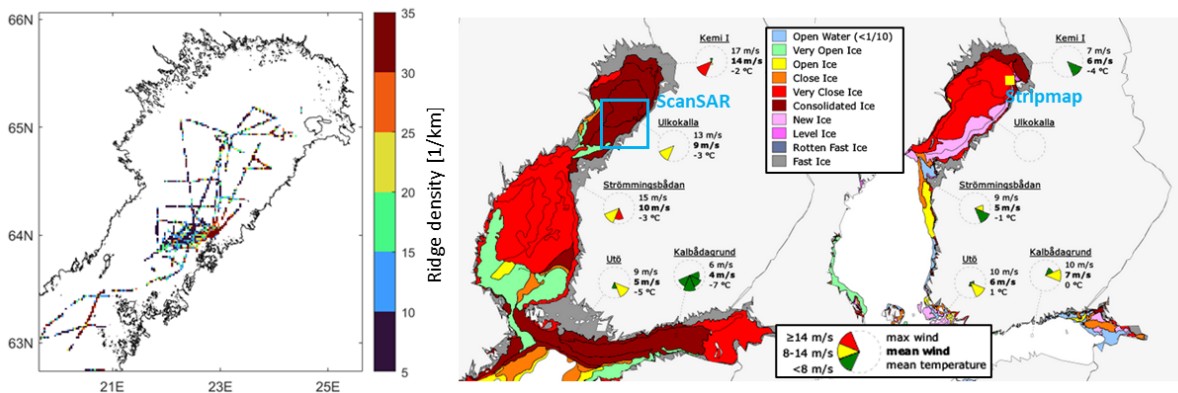

**Figure 1.** The profiling measurement lines during the 2011 field campaign (left panel), location of ScanSAR image with prevailed ice
conditions (middle panel), location of Stripmap image and ice conditions (right panel).

The analysis of laser data followed standard procedures described in (Eicken et al., 2009). The zero level was determined by
conducting a two-step high-pass filtering with minimum point selection between the steps. Then local maxima were identified
by Rayleigh criterion demanding that for two successive maxima the minimum elevation between them must be less than half
from either maximum. Otherwise the shallower one is not counted. The elevation distribution of Rayleigh separated maxima
has negative exponential tail for values higher than 0.4 m that was selected as cutoff elevation. Above the cutoff the maxima
are assumed to be predominantly ridge sails. The average ridge height and density are 0.65 m and 11.7 1/km.





### 4.3 Weather and Ice Conditions

#### 4.3.1 Winter 2010/2011

The winter 2010/2011 prior the field campaign was colder than average. When the field campaign to the Bay of Bothnia started in late February, the the ice extent was larger than average annual maximum extent (Figure 1) . There was a drifting ice station in the southern part of the basin from 25 February to 4 March (initial location 63°45.85' N,l21°55.34' E) after which the helicopter campaign continued to 7 March. During this period the Bay of Bothnia did not have 100% ice concentration because high SW winds, the mean wind speed being 10 m/s with variation from 5 m/s to 18 m/s, repeatedly triggered drift and deformation periods and opening in the southern part of the basin. The wind direction changed NW during the helicopter campaign. According to the Finnish Ice Service ice charts the level ice thickness near the coast varied from 30 to 70 cm and from 30 to 60 cm in the middle of the Sea of Bothnia. The modal thicknesses of the HEM profiles are mostly within this range as well.

The air temperature began to increase from below -10 C° (on 24 February) to -1.5 C° (27 February). When the TSX image was acquired $T_a$ was -2.3 C° in the ice station.

Short snow lines were measured during 2-4 March. The mean thickness was 8 cm and the standard deviation (STD) 11 cm. About 50 cm thick snow accumulations were often found by ridges, sometimes covering the ridge sail. On 3 March the snow had already some moisture and the density varied 0.2-0.3 kg/m3 on level ice and 0.3-0.4 kg/m3 near ice ridges. We can assume that during the TSX image acquisition the snow was still dry and it did not affect significantly the backscattering.

The ice station drifted about 0.1-0.4 knots during the time gap between the TSX image and the helicopter flights. Throughout this time period additional deformation occurred in the area where the flights were performed. Hence neither the location nor the deformation characteristics of ridge fields stayed as they were at the time of TSX image acquisition. When assessing the results, these changes must be taken into account.

#### 4.3.2 Winter 2015/2016

The winter in 2016 was mild and only the Bay of Bothnia had ice concentration of 100%. Recurrent periods of mostly SW winds induced cycles of deformation, opening and freezeup and only in the beginning of March the basin attained more persistent ice cover consisting of ridged and rafted ice types. On 5 March (Figure 1) fast ice thickness in the NE quadrant of the basin was 50-65 cm, the level ice thickness in the ridged ice pack 30-50 cm, and the air temperature from -1 to -4 degrees. The temperature stayed below zero and no snowfall occurred during 10 days before the Stripmap SAR image acquisition date, the snow thickness on mainland being about 30 cm.

# 5    High resolution SAR image analysis

## 5.1    Ridging signatures and contextual images

The pixel size of the Stripmap image is 1.25 m which resolves backscattering signatures from individual ridges. The size of the image (Figure 2) is 19x19 km but for analysis and visualisation we used also different subimages, principally upper left 7000x7000 (8.7x8.7 km) subimage excluding the refrozen lead and a 1024x1024 subimage (Figure 4) with representative ridging signature.

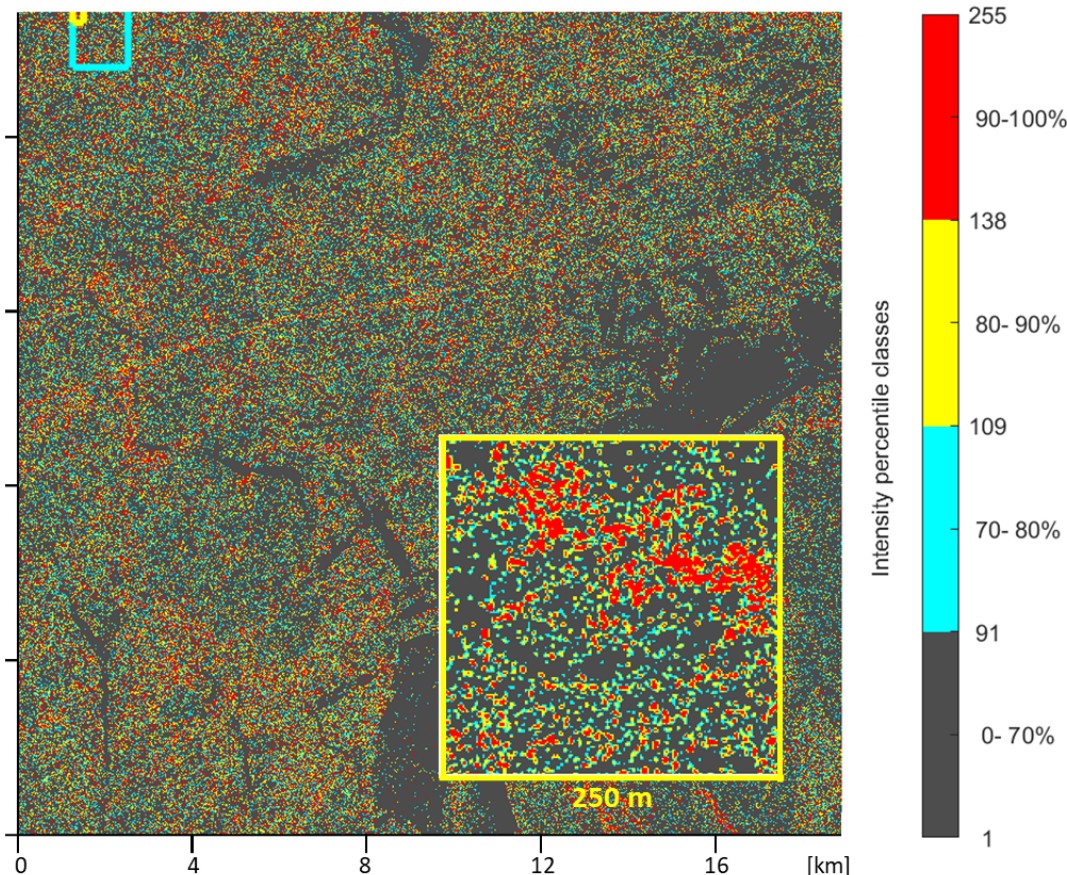

**Figure 2.** High resolution 19x19 km Stripmap image presented in terms of percentile classes. The location of the 1024x1024 subimage (Figure 4, upper left panel) is indicated, as well as the location of futher 200x200 closeup shown also as an insert.

For better visualisation, images are presented in terms of four intensity percentile classes so that the brightest 30% of pixels is divided into three 10% wide classes. In the 1024x1024 subimage (Figure 4 upper left panel) ridging features appear then as delineated by the brightest 20 percent while the range 70-80% begins to add more scattered pixels. The percentiles of the




1024x1024 subimage were 105 (-17.7 dB) for 70% , 124 (-15.5 dB) for 80%, and 159 (-11.3 dB) for 90%. It is seen that the hope of a similar quasi-photorealistic appearance as is found in comparable TerraSAR-X terrestrial images by Dumitru and Datcu (2013) fails to materialise. There are obvious linear ridges but these consist of chains of detached bright components that do not connect into continuous features. Larger features, reminiscent of ridge groups or small rubble fields, are more

readily observed. However, they are also aggregates of detached bright components, with a texture not much different from ship channels that can be assumed to be flat and have uniform spatial distribution of bright scatterers. Increasing the bright pixel percentage from 30% improves slightly the connectedness of ridge signatures but at the same time obscures their delineation by adding scattered bright pixels in between.

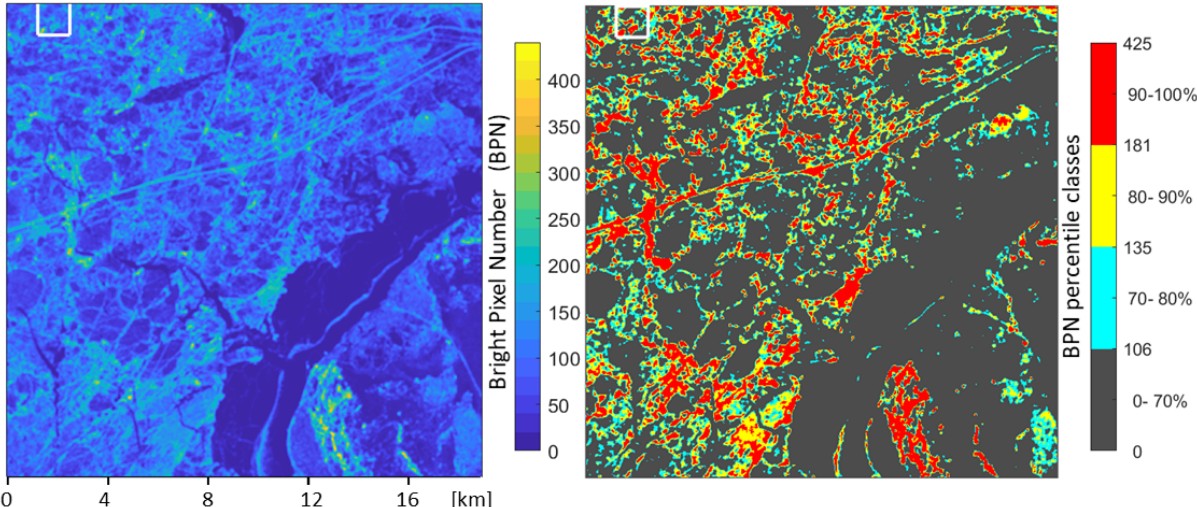

**Figure 3.** The contextual image derived from the full image with BPP 20% and L=121, and its presentation in terms of percentile classes.

These observations agree with the results by Manninen (1992) and Carlström and Ulander (1995) indicating that bright
returns from ridge rubble depend on random factors like favorable orientation of blocks while the returns from the remaining sail may fail to be clearly distinct from the surrounding level ice. The apparent randomness also corroborates the assumption that the density of the bright pixels in the image is proportional to coverage of ridge rubble and may act as proxy of ridge rubble coverage and ridge density.

To proceed, following the approach outlined in Section 3, a certain bright pixel percentage (BPP) is selected and a binary
image with unit values for the bright pixels is generated. The variation of bright pixel density is described in terms of bright pixel numbers (BPN) in pixel blocks with variable side length $L$, or equivalently in terms of bright pixel densities BPN/$L^2$ in scale $L$. This can be done in terms of a non-overlapping block tiling, which is preferred in the statistical context of Section 6, or as in the present context in terms of a sliding operation for the binary image, accomplished by a convolution with $L$x$L$ unit





matrix kernel and with required adjustments near the boundaries when the kernel does not fit inside the image. The result of the sliding operation is 'contextual image' with the same size as the original image.

Suggested by Figure 2, the BPP is chosen to be 20%. The block side lenghts $L = 21$ (26 m) and $L = 101$ (126 m) are selected for the 1024x1024 and full images respectively. The contextual images are also shown as percentile class images where the
30% of highest BPN values are divided into three 10% wide classes. It is seen that the ridging signatures are enhanced and their connectedness is improved in the contextual images (Figure 3, Figure 4 upper right panel).

Including a pragmatic viewpoint, $L = 21$ and $L = 101$ correspond to the width and length scales of icebreaking ships so that the BPN values of contextual images provide the number of bright scatterers the ship bow or the whole ship interacts at a time at the pixel location. If the BPN is accepted as a proxy of ridge rubble coverage, the contextual image has direct pertinence to
the navigability. Efficient ship route optimisation in ridged ice cover is based on engineering models for the added resistance from the ridges. The ice-going speed and the probability of besetting can then be obtained in simulations that take as input the ice conditions along alternative tracks (Kuuliala et al., 2016). The contextual image has potential to provide data for the purpose. The other possibility is to use ships' remotely observed response to classify the contextual image pixel values in terms of ice cover resistance (Similä and Lensu, 2018). The percentile classes of contextual images can then be interpreted as classes
of navigational difficulty, or rather as delineation of areas that a ship should not enter. They can be used in tactical navigation or in route optimisation that only seeks avoid difficult ice types.

## 5.2   Sensitivity of contextual images

The pixel block side length $L$ affects the resolution of detail in the contextual images and its value may be chosen to suit the context; hence the term. The question on the effect of BPP to the contextual images is of principal importance. There are also
alternative methods to generate a contextual image, like sliding block average for intensity. With increasing BPP the variation of the BPN in the contextual image, measured by the coefficient of variation, decreases towards zero and the contrast between ridged areas and the background grows weaker. However, this appears less in the percentile class images. Choosing BPP from the range from 2% (pixels with value 255) to 95% results into a percentile class images that are not much different from that in Figure 3. This is also found if sliding intensity average is used instead of BPP. More strikingly, the information of the contextual
image is essentially retained in different randomising transformations done for the original image. In Figure 4 (lower left panel) the 1024x1024 image has been multiplied eight times by a random matrix which erodes the ridging signature of the original image almost beyond recognition. However, the contextual images still emerge essentially unchanged in their main features (lower right panel).

These observations indicate that, within the considered pixel block methods using the same block size, the order of the BPN
values is only weakly affected by method change. For the 10% class widths of the percentile class images the change does not induce extensive flux of contextual image values across class boundaries. In more detail, starting with some reference method it can be investigated how the contextual image values are mapped to those of an alternative method. For example, mapping for $L = 21$ and BPP 20% the 1024x1024 image BPN values to matching $L = 21$ sliding intensity averages the relationship is monotonous on the average. For each BPN value the mapped intensity averages have approximately a normal distribution







**Figure 4.** The 1024x1024 subimage (upper panel, left) multiplied 8 times by a random matrix (lower panel, left). Contextual images $L = 21$ for both on their right hand side.

with standard deviation in the range 3.5-4.5. On the other hand, the widths of the $70\% - 80\%$, $80\% - 90\%$ and $90\% - 100\%$ percentile classes for the intensity average image are 9, 15 and 102 respectively. Thus especially the $90\% - 100\%$ class retains its delineation in this method change.

In the present study the BPN statistics is at focus however. For selected pair of $L$ and BPP a certain finite, discrete distribution with maximum value $L^2$ and mean value $\text{BPP} * L^2/100$ becomes defined for the BPN. A distribution model parameterised by $L$ and BPP would provide basis for the study of interrelations between contextual images defined for different scales $L$ and BPP values. The principal approach towards the BPN statistics is to consider how the statistics is changed in BPP changes.

As an introduction to the statistical approach, the BPN values are obtained from a non-overlapping $L = 8$ block tiling (i.e. not as a sliding operation of contextual images) for the 1024x1024 image. The BPP is increased from 10% to 15%. For each block this induces a BPN increase from $n$ to $n + m(n)$ where $m(n)$ is the number of added bright pixels. In Figure 5 the range of $n + m(n)$ is shown for each value $n$ together with the distribution $m(n)$ given $n = 11$. The mean value of $m(n)$ increases

first with $n$, but levels off in the mid-range and starts to decrease for higher BPN. For $n = 11$ and other selected the test values $m(n)$ follows Poisson distribution. Thus the change $m(n)$ can be modeled as resulting from a process where new bright pixels are assigned randomly to the blocks with a rate that depends on $n$. This observation leads to the generative model for the BPN distributions in the next section.

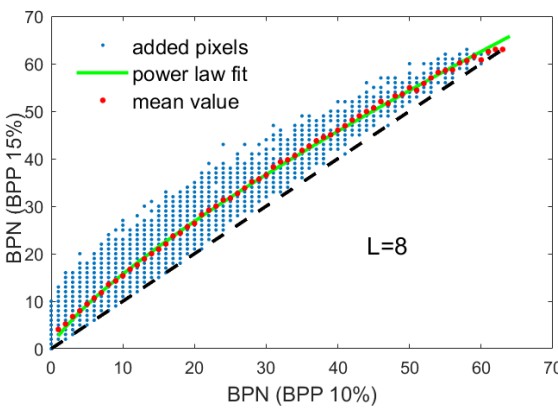
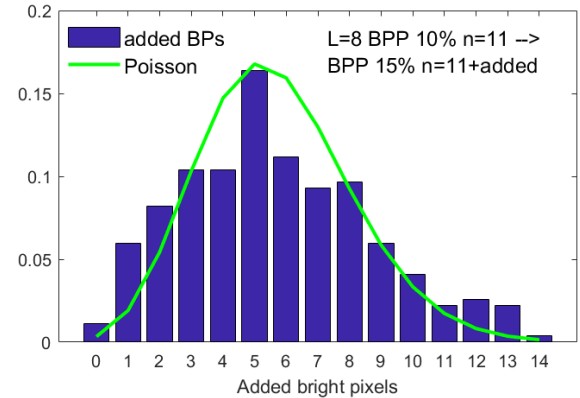

**Figure 5.** The increase in BPN values for $L = 8$ blocks when BPP is increased from 10% to 15%. On the left, the range and mean value of increased BPN. On the right, distribution of BPN increase for blocks with 11 bright pixels.

## 6   Statistical model

### 6.1   Ridge sail rubble, ridge density and ridge sail numbers

In the present approach the statistics of SAR images from ridged ice cover is described in terms of bright pixel numbers (BPN) in non-overlapping pixel blocks. High resolution ( 1.25 m) images resolving individual ridge sails are used in the development. Bright pixels are defined by an intensity threshold, or by equivalent bright pixel percentage (BPP), and the numbers of bright pixels in the pixel blocks are counted. The parameters of the approach are the block side length $L$ in pixels and the BPP

(equivalently, intensity threshold for the BPP).

The BPN is preferred to the bright pixel density $\text{BPN}/L^2$ as it is integer valued additive measure of the image and finite, discrete statistics can be used. The additive measures and corresponding densities are interchangeable provided that the reference to the scale $L$ is respected. The guiding hypothesis is that BPN values increase with the area covered by ridge sail rubble which is the principal ground truth quantity. The corresponding ground truth density is (relative) coverage of ridge sail rubble





in scale $L$. It is less evident whether or not the BPN values increase with ridge height if the sail rubble area remains the same. This cannot be answered with the present ground truth data, but the observation that ship channels and rubble fields appear not much differently in the high resolution image suggests that the height has at most minor effect.

Ridge sail rubble can be quantified from two-dimensional topographic data but only few local datasets exist from the Baltic, one analysed data being (Similä et al., 2010). The more extensive data consists of airborne ice surface profiles from which ridge sails and their heights are identified. Profile data will retain its importance also in the future as narrow beam satellite laser altimetry by IceSAT$-$2 is likely to become a major source of validation data (Fredensborg Hansen et al., 2021). For pencil beam airborne laser data, which is used in this study, ridge sail statistics is described in terms of profile segments of length $L$. The numbers of ridge sails reaching above certain height threshold in the segments are counted (ridge sail numbers, RSN). Segment length $L$ and the sail height threshold are parameters. They are conceptually equivalent to the pixel block side length and the pixel intensity threshold for SAR images. Ridge density RSN/$L$ in units 1/km is here a scale dependent distributed quantity while in literature 'ridge density' usually refers to the average value of an extended surface profile.

In the Section 7.1 we will derive Equation 7, which yields for the average Baltic conditions an estimate that unit ridge density corresponds approximately to 2% sail rubble coverage. The crucial step in the estimation is to derive the total length of ridge sail per unit area from observed ridge density. To apply this step to a specific $L$x$L$ area in deterministic fashion requires either that the profile data crosses the area many times along different tracks and directions, or that the area is very large and has isotropic and homogenous ridging conditions. Otherwise only a correlation can be expected. The same applies then to the relation between observed RSN and BPN values even for spatially matching data. In addition, due to temporal separation and ice drift, BPN values in pixel blocks and RSN values in segments cannot usually be directly compared for matched pairs of blocks and segments. The present approach therefore proceeds purely statistically, seeking to demonstrate that both BPN and RSN follow the same statistical model, and to establish parametric relations between the two.

## 6.2 Threshold process

For an idealised SAR image with accurate real-valued intensities the pixel values can be arranged into strictly increasing order. The derivation of the statistical model for the BPN in pixel blocks is based on the following approach. In the idealised setting, starting from empty image matrix and from the brightest pixel, the pixels of the ordered series are added one by one to the matrix. It can then be studied how the probability for the BPN to increase by one depends on the state <L,BPN,...> of the block as defined by side length $L$, BPN, and possibly by other descriptors. These increase probabilities can then be used to formulate recurrence relations that can be iterated to generate finite BPN distributions. This is analytically manageable at least when the relations depend linearly on BPN, which will be also the assumption behind the distribution models derived below.

For integer valued SAR images the process is realised by starting from the BPN values for maximum intensity, usually 255, and adding step by step pixels of subsequent lower intensity. This means decreasing the intensity threshold by unit integer steps, and the process is designated here as threshold process. The increase probabilities are replaced by increase rates, that is, the relative number of events of unit BPN increase. For RSN an analogical process decreases sail height threshold with small values, starting from a threshold equalling the highest ridge. In either case the increase rates are then interpreted as increase





probabilities of an idealised process. Strictly taken this requires that the number of increase events larger than unity is relatively small for each step. In practice the steps are limited from below by the discreteness of the SAR intensity, or by the nominal reading accuracy of profiling instruments, typically 1 cm.

The threshold process is continued to certain BPP or ridge sail height cutoff, called a target threshold above which the

statistics is assumed to be dominated by ridges. The derivation of statistical model from the increase rate is based on the following observation. The target threshold being set, the SAR intensity or ridge sail height values higher than or equal to the target threshold can be randomly permuted. This changes the increase rates, generating another threshold process. However, the end result, or the distribution observed on the target threshold level, is the same for both processes. Thus the increase rates for the randomly permeted case can be used to generate the observed distribution. The random permutation has the merit that

it removes the spatial correlations of intensities or sail heights. The threshold process reduces to a random deposition process that does not depend on intensities or sail heights.

### 6.3   Scale system of distributions

The threshold process and the distribution model are presented for ridge sail numbers. The case for bright pixel numbers is conceptually analogical. However, the BPN statistics involves several features present in SAR images but not in profile data:

discreteness of integer intensities, resolution set by pixel size, and the limit $L^2$ as the maximum BPN value. Especially for small $L$ and high BPP the saturation of the blocks affects the threshold process significantly.

In RSN analysis surface profile data is divided into segments and the numbers $n$ of sails in the segments are counted. Segment length $L_i$ is a variable parameter. A discrete distribution $k(n_i)$ for sail number $n_i$ becomes defined for each scale $L_i$. The mean value is related to the average profile ridge density $d$ as $<n_i> = dL_i$. If $L_j < L_i$ is a shorter segment, a conditional

distribution $k(n_j|n_i)$ becomes defined. This can be interpreted as the conditional probability to find $n_j$ sails in an $L_j$-segment nested inside an $L_i$-segment containing $n_i$ sails. The $k(n_j|n_i)$ can be also called downscaling probabilities as they can be used to derive distribution $k(n_j)$ from $k(n_i)$. This approach can be extended to a cascade of scales, constituting a scale system of distributions. In that case it is convenient if shorter scales divide longer scales, preferably as a binary cascade where segments are successively bisected.

In an idealised threshold processes the sail heights can be arranged into strictly increasing order and added one by one to the segmented profile. In the following two nested segments with lengths $L_i > L_j$ are considered so that the longer segment $L_i$ is divided into subsegments with length $L_j$ and $L_i - L_j$. If in the process steps the sail becomes added in the process to the $L_i$-segment, it has a certain probability to become added to each subsegment. The associated probabilities must satisfy the additivity condition $P(L_i) = P(L_j) + P(L_i - L_j)$. In general the probabilities depend on local conditions, especially sail

height correlations. If the correlations are removed by a random permutation of heights the simplest assumption satisfying the additivity condition is $P(L_i) \sim n_i + aL_i$. This assumption leads by iteration to the hypergeometric distribution (Lensu, 2003):

$$k(n_j|n_i) = \frac{\binom{aL_j+n_j-1}{n_j}\binom{aL_i-aL_j-1+n_i-nj}{n_i-n_j}}{\binom{aL_i+n_i-1}{n_i}}, \quad n_j = 0, 1, \ldots, n_i. \tag{1}$$



If $L_i/L_j \gg 1$, then the finite and discrete distribution $k(n_j|n_i)$ is approximated by the negative binomial distribution

$$k(n_j) = \binom{aL_j + n_j - 1}{n_j} p^{aL_j}(1-p)^{n_j}, \quad \text{where} \quad p = \frac{aL_j}{aL_j + <n_j>} = \frac{a}{a+d}, \tag{2}$$

defined for all integer values $n_j$. Here $d$ is the average ridge density. Further, if mean value $<n_j>$ is large, the continuous approximation of Eq. 2 is the gamma distribution:

$$f(n_j) = \frac{1}{\Gamma(\alpha)} \beta^\alpha n_j^{\alpha-1} e^{\beta n_j}, \quad \alpha = aL_j, \quad \beta = \frac{aL_j}{<n_j>} = \frac{a}{d}. \tag{3}$$

However, it is convenient to apply the given system so that regional scale $L$ has ridge density $d$ and the variation in ridging conditions is described by the negative binomial $k(n_i)$ as approximated by the gamma distribution. The hypergeometric $k(n_j|n_i)$ can then describe local variation or be used as downscaling probabilities.

The scale system is validated for both RSN and BPN statistics along in two stages. The additivity condition is first validated by conducting a threshold process for selected scales and target thresholds and observing whether the increase rates have linear dependence on the RSN and BPN. In the next stage, the distribution models are fitted to the data. In the sections to follow this is made by observed mean and variance and the goodness of fit is checked. The parameter $a$ is obtained from the variance

$$var(n_j|n_i) = \frac{L_j n_i (aL_i + n_i)(L_i - L_j)}{L_i^2(aL_i + 1)} \tag{4}$$

and

$$var(n_j) = \langle n_j \rangle + \frac{\langle n_j \rangle^2}{aL_j} = dL_j \frac{a+d}{a} \tag{5}$$

for the hypergeometric and negative binomial models, respectively. We remark that the variance for the negative binomial distribution is larger than the mean which ensures that $a$ is positive in Eq. 5. The parameter $a$ quantifies in the ridge sail threshold process the relative strength of component process of random spatial deposition of sails, that is, Poisson process. In particular, non-zero $a$ is required for the initial transition of RSN or BPN from zero to one in the threshold process. If the validation results agree for test cases selected from a certain ranges of scales and target threshold values , it can be concluded that the scale system is applicable above the lowest target threshold, over the whole scale range, and for any combination of scales within the scale range.

### 6.4 Observed rates and distributions for ridge sail number

The profile data set described in Section 4.2 was used to study the scale system and its generative threshold process. All profile data consisting of profiles exceeding 5 km in length was selected and truncated to be divisible into 1.6 km, in total 2256 km. In the determination of RSN values for variable segment lengths $L$ the binary cascade [50,100,200,400,800] (meters) was used. The threshold process was set to decrease in 0.01 m steps from 2.5 meters, which value selects 10 ridge sails from the whole dataset. The weight of cases where RSN values increase by more than one in a step increases exponentially with the decreasing threshold. This affects the results significantly for thresholds smaller than 0.5 m. The target threshold was set therefore to 0.5 m, selecting 18638 ridges, and the number of threshold process steps was 201.





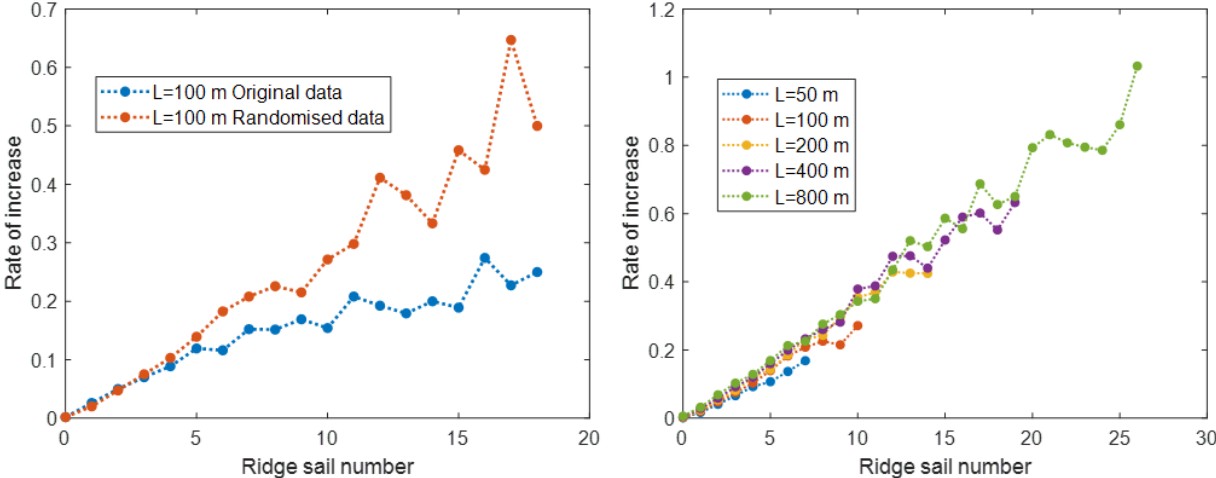

**Figure 6.** Sail number increase rate for in a threshold process with target threshold 0.5 m. Left: Rate for observed sail heights and randomised sail heights for segment length $L = 100$ m . Right: Rates for randomised sail heights for different segment lengths.

The increase rates were determined as the mean value of RSN increases, including zero increase. The presented results are for the pooled data comprising all steps to the 0.5 m target threshold. However, the results were essentially similar if a higher target threshold was used. The increase rates were calculated both for the original data and for the data with randomly permuted sail heights. The results are shown in Figure 6 where the graphs are truncated for the largest RSN values for which the few
RSN increase instances show excessive variation. In the right panel of Figure 6 the highest RSN already corresponds to ridge density 160/km, a value characteristic to rubble fields with 100% ridge rubble coverage. The increase rates were generally linear for the randomly permuted data. There is a slight superlinear bend when both $L$ and RSN have higher values.

The increase rates for the original, non randomised data are shown also in Figure 6 for $L = 100$ and are seen to deviate after beginning from linear to sublinear increase. For other scales the behaviour was similar and for long $L$ the rate more or
less settled to a constant value in its tail part. Sail height correlations are the likely reason for these observations. These arise naturally as the thickness of parent ice from which the ridges are created varies spatially and during the course of ice season. Clear correlations were also found between RSN and the average sail height for a segment.

The hypergeometric and negative binomial distributions were fitted to data with observed parameters (mean and variance). The negative binomial $k(n_i)$ agrees well with the empirical distributions derived from the full dataset in Figure 7. The mean
value for each negtive binomial case is $dLi$ where $d = 11.7$ is the average ridge density for the whole dataset and variance as in Equation **??**,. Only the number of empty segments is overestimated for 200 m and 400 m segment lengths. Figure 8 shows the hypergeometric distributions $k(n_j|n_i)$ for the conditioning scale $L_i = 1600$ m and for the range [50,100,200,400] of subsegment scale $L_j$. Results for two values of conditioning RSN, $n_i = 16$ and $n_i = 72$, are shown. The mean value for each hypergeometric case is $n_i L_j / L_i$ and the variance as in Equation 4,. These are equivalent to ridge densities 10/km and 45/km


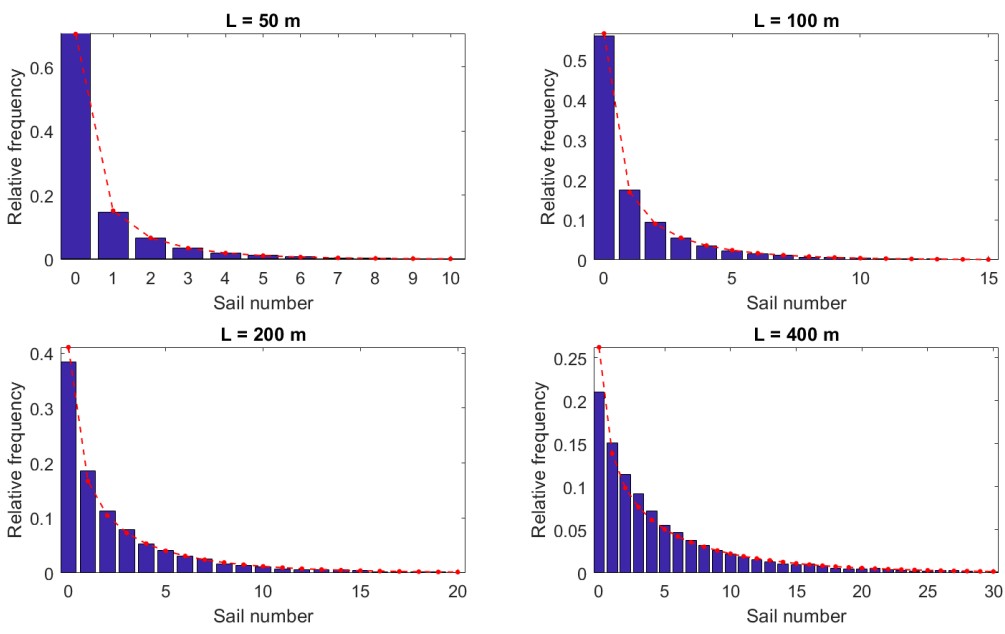

**Figure 7.** Negative binomial fits to the RSN distribution for the full dataset and for different segment lengths.

for the segments $L_i$. The agreement is good considering that the subset size is limited by fixed value $n_i$. Similar results are found for other combinations of $L_i, L_j,$ and $n_i$.

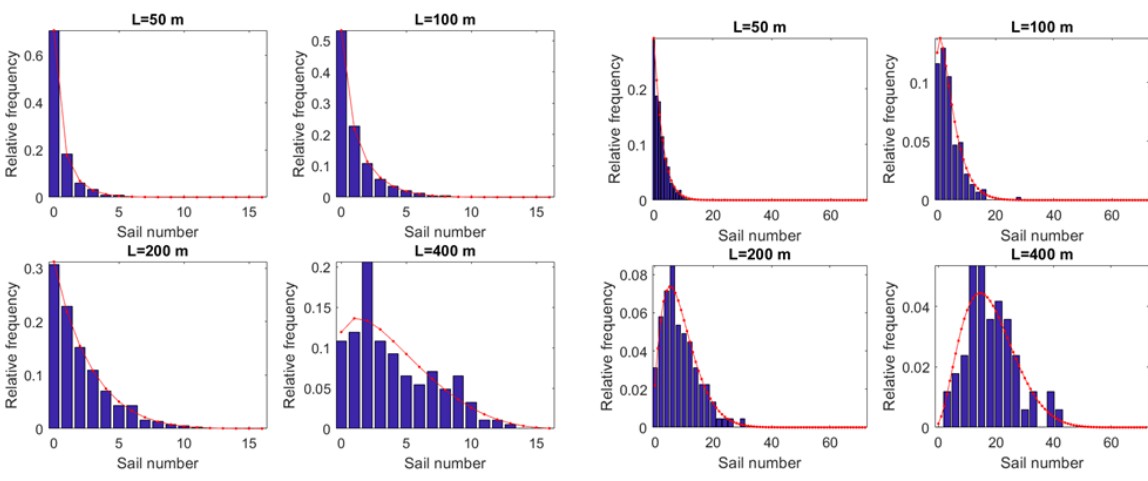

**Figure 8.** Hypergeometric fits for distributions $k(n_j|n_i)$ conditioned by $L_i = 1600$ m and $n_i = 16$ (left panel group) and $n_i = 72$ (right panel group). The scales $L_i$ belong to [50,100,200,400].





The hypergeometric model was derived from the assumption that the rate of RSN increase for segment length is proportional to $n + aL$. The parameter $a$ is obtained from mean and variance and is generally found to decrease with $L$ following a power law. For the shown hypergeometric case the exponent has about the same value 0.5 for all values of the conditioning $n_i$ ranging from 16 to 72. The parameter $a$ was interpreted as relative Poisson intensity that also provides the rate of sail appearances to

empty segments in the threshold process. The threshold process is basically governed by the physical presence of ridge rubble in the profile segments. In an idealised setting, if there is not rubble present it cannot be detected by the threshold process either. Thus the power law suggests that the area covered by ridge rubble has fractal geometry. This point is addressed further in the Discussion section.

### 6.5    Observed rates and distributions for BPN

For the BPN threshold process analyses the upper left $7000 \times 7000$ subimage of the high resolution image (Figure 2)was used. Instead of contextual image generated by a convolution operation, the image was divided into non-overlapping pixel blocks with varying side length $L$, measured in pixels. The intensity threshold was decreased by unit integer steps, starting from 255, and the rate of BPN increase as a function of BPN was obtained for each step. In the presented results the rate data comprises all steps down to the target threshold. In the subimage 1.2% of the highest intensities are saturated to 255. This percentage

for the single value 255 is equal to that for the adjacent intensity band [218,254] and would correspond to the extending of the exponential tail of intensity histogram beyond 255 to 350. This lacking intensity resolution for the brightest scatterers somewhat affects the results for the initial steps of the threshold cascade.


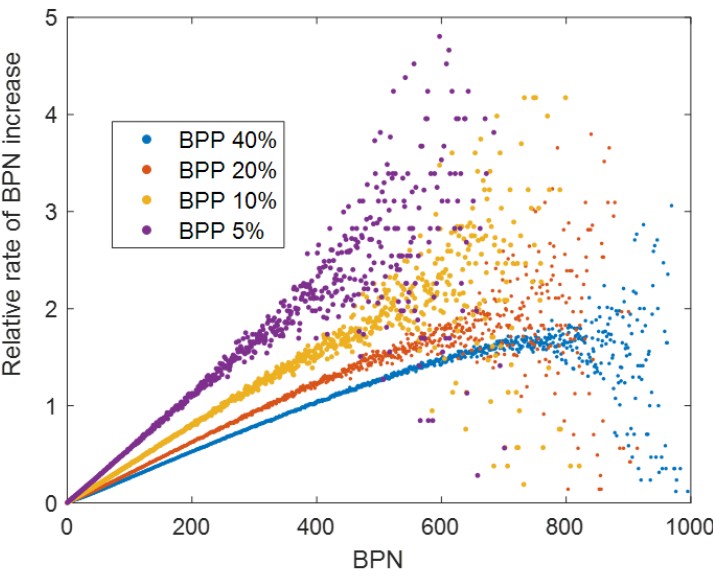

**Figure 9.** The BPN increase rates for randomised threshold process. Pixel block side length $L = 36$ and BPP increases from 5% to 40% in a binary fashion.

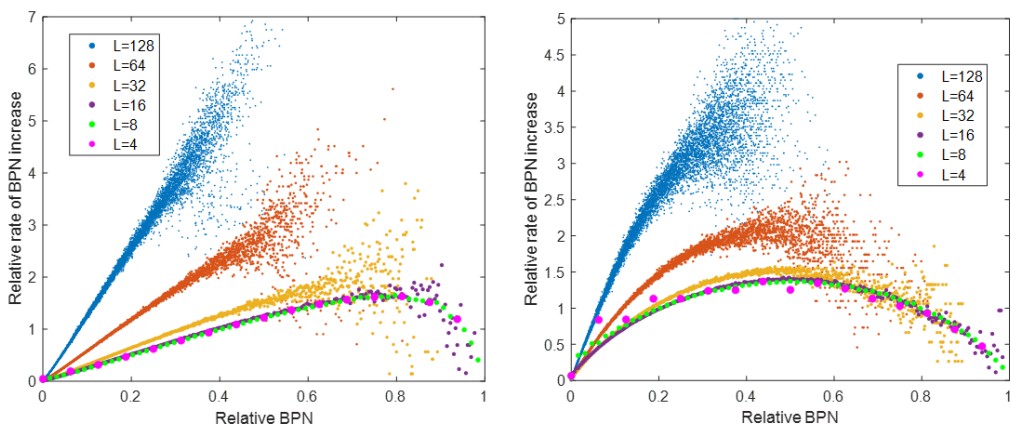

**Figure 10.** The BPN increas rates for BPP 20% and for pixel block side length increasing from 4 to 128 in binary fashion. The left panel is for randomised threshold process, the right panel for non-randomised process.

Analogically to the RSN analysis the pixel intensities above and including the target threshold were randomly permuted. The increase rate was investigated first for a fixed block side length $L$ and varying target thresholds expressed as BPP. Then





the same was done for varying block side length and for a fixed target threshold. The first test case is presented for $L = 36$ in Figure 9, where the BPP values [5,10,20,40] correspond to intensity thresholds [182,149,118,86]. Similarly as for the RSN the absolute magnitude of the rates is not relevant, only their possible linearity. For presentation purposes the rates have been scaled so that the sum rate equals $L^2$ or 1024. The different degrees of saturation then separate the point clouds in Figure 9.

Linear dependence of rates on BPN is observed unless the saturation of the pixel block is felt. This occurs after 3/4 of the block capacity is filled and affects significantly only cases BPP 20% and 40%.

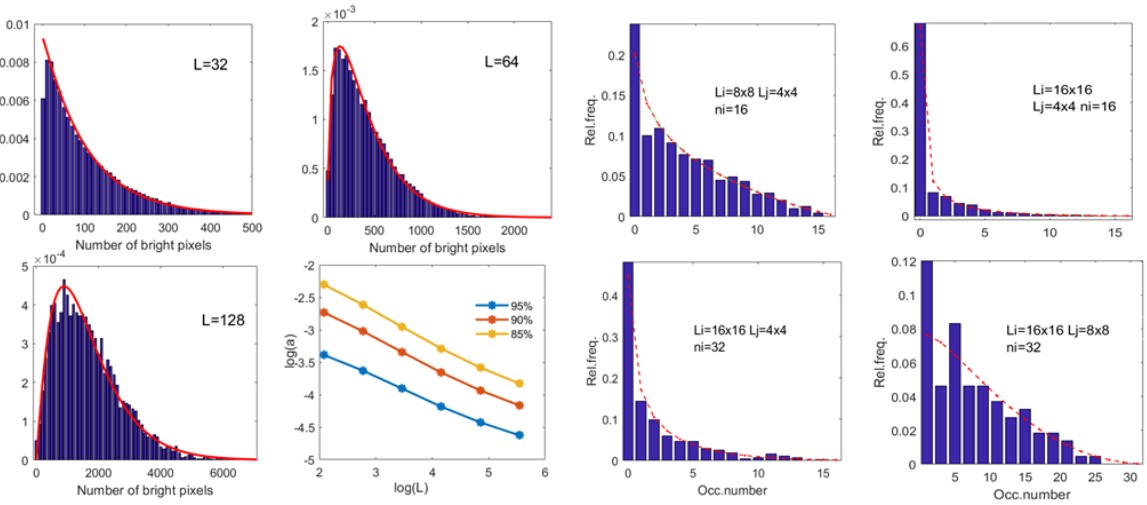

**Figure 11.** In the left panels BPN distributions and their negative binomial fits for BPP 10% and three different pixel block side lengths $L$. The parameter $a$ is also shown for a scale cascade $8, 16, \ldots, 256$ and for three different BPP. In the right panel hypergeometric fits for conditional BPN distributions for different combinations of conditioning scale $L_i$ and conditioning BPN $n_i$.

The other test case is presented for BPP 80% in Figure 10, where the block side length increases in binary fashion from 4 to 128. In addtion to the scaling of rates described above the BPN range $[0, L^2]$ has been scaled to [0,1]. The results are similar as above. For comparison, the Figure shows also the same result for the original non-randomised. For higher $L$ the relationship

is approximately linear for small relative BPN values, but otherwise it is quadratic and the second order fit is also symmetric within the range of nonzero rates. Intensity correlations are the apparent reason but no further conclusions are attempted. Also for the variable BPP, constant $L = 36$ case (not shown) the quadratic behavior is found for nonrandomised data.

The agreement of the proposed distribution system with the data is again good when fitting with mean and variance. For negative binomial test cases the mean value is BPP$*L_i^2/100$ and for the hypergeometric test cases $n_i L_j / L_i$ while the variances

are in Equations 5 and 4. To maximize data amounts the empirical distributions were calculated from block tiling of the $9216 \times 9216$ ($11.5 \times 11.5$ km) upper left corner subimage of (Figure 2). The dimension is largest divisible by 1024 and excluding the refrozen lead. Figure 11 shows negative binomial distributions $k(n_i)$ fitted by observed mean and variance. BPP is 10% and the results for three scales are shown. The fourth subplot shows the model parameter $a$ as obtained from mean and variance





for the binary cascade of scales $L_i = 8, 16, \ldots, 256$, and for three BPP values. The power law exponents varies from -0.37 to -0.45 and are thus close to values obtained in RSN analysis. Also the negative hypergeometric model $k(n_j|n_i)$ conditioned by scale $L_i$ and BPN value $n_i$ applies well. Figure 11 show results for selected combinations of $Li, Lj$ and $n_i$. The conditioning scale cannot exceed much the value $L_i = 16$ as the number of instances $n_j$ for conditioning pairs $(L_i, n_i)$ decrease as $L^{-4}$.

## 6.6 Simulation of SAR texture by the threshold process

The hypergeometric distribution model was derived from the assumption that the RSN/BPN increase rate of the threshold process has linear dependence on RSN or BPN. This assumption was validated by the rates calculated from data and by fitting the model to data by mean and variance. To analyse the increase probabilities of the threshold process in fullest detail would require that ridge sails or bright pixels are added one by one by process steps. For present integer valued images this cannot be attained. On the other hand, it is possible to test generative hypotheses by simulating the corresponding generative process. The simulation commences from a certain stage of the observed threshold process. New objects, here bright pixels or sails, are added one by one following the assumed transition probabilities. The simulated process is then compared with the observed one. This allows also the analysis of the spatial aspects of the generative process, most importantly homogeneity and support, or the subregion outside which the process is not active.

This approach is formulated here for SAR images using the linear assumption but including certain additional restrictions related to the spatial distribution. A binary image with value 1 for low percentage of brightest pixels is used to seed the process. The simulation changes the value of zero pixels into one, one at a time. This is done by a scale cascade following the linear generative hypothesis. The first step divides the image into four rectangles along randomly chosen vertical and horizontal lines. To each rectangle is assigned weight $aN0 + N1$, where $N0$ is number of pixels, $N1$ number of occupied pixels, and $a < 1$ is a parameter that controls the relative strength of random spatial placement (Poisson process). One of the rectangles is selected for next step with a probability defined as the ratio of rectangle weight to the sum of weights. The selected rectangle is divided further into four and the cascade is continued until there is only one non-occupied pixel in the rectangle. The status of this pixel is changed to one and the process repeats until a preset number of nonzero pixels is reached.

If $aN0 \gg N1$, Poisson process dominates the simulation step, but overall the process inevitably gravitates towards image areas with higher density of nonzero pixels. However, if the fraction of nonzero pixels is initially low, the Poisson component process tends to add nonzero pixels into larger empty areas, creating first dispersion and then spurious clusters. To reduce this, an additional condition requires that the nonzero pixel cannot be added within a rectangle if its fraction of nonzero pixels is below a threshold $Co$ and the rectangle size is within certain limits $[N1, N2]$. This effectively means restricting the support of the process.

The simulation process was applied to the same 1024x1024 image as in Figure 3. The initial binary image corresponds to BPP 3% . The simulation added 10% so that the end result BPP is 13%. The parameters were $a = 1/20$, $Co = 1/100$, and $[N1, N2] = [100, 20000]$ and the results are shown in Figure 12 Although not matching exactly on pixel level both 13% images exhibit the same pixel density variations. Convolving with $21 \times 21$ kernel the and generating contextual and percentile class images similar to those shown in Figure 3 the results for the real and simulated 13% images match almost exactly, that

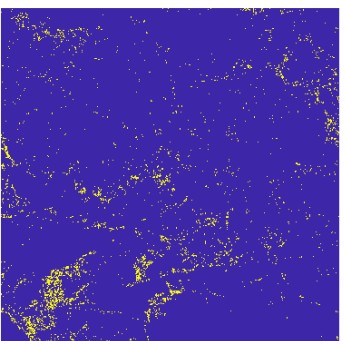 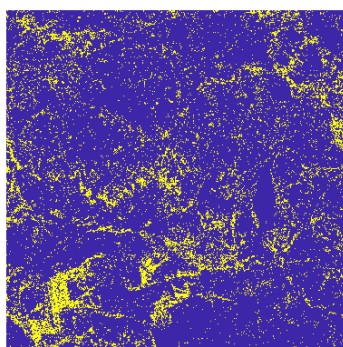 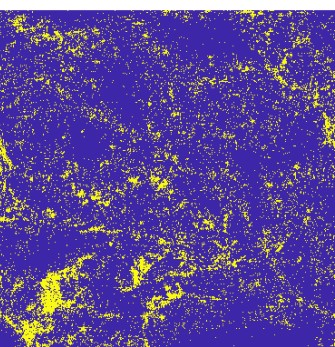

**Figure 12.** BPP 3% binary image, BPP 13% image, and the BPP 13% image generated from the BPP3% image by the simulation algorithm.

is, they contain the same information at scale $L = 21$. The results provide additional corroboration for the soundness of the generative hypothesis. They provide also insight that certain parts of 'non deformed' ice should be left outside the ridging statistics, a feature suggested also previously by the emergence of power law exponents for the parameter $a$ in the distribution fits. The parameters of this demonstration were determined by simple experimentation, but more systematic approaches can be

conceived. Especially, the intensity values could be incorporated which would introduce spatial intensity correlations.

# 7  Data analysis for medium resolution SAR data

## 7.1  SAR and LASER comparison for 2011 image

To analyse ridged areas in the TSX ScanSAR image of resolution 20 m (Fig. 13, left panel, see also Figure 1) we proceed in a similar manner than in the analysis of the high resolution SAR data above. The image size 106 x 94 km subimage ($5300 \times 4700$

pixels) was chosen to match nearly simultaneous laser profile data.Because we have an access to actual laser profile data, this section can be considered as validation of the analysis method introduced in Sections 5 and 6. The analysis is performed using ridge and rubble coverage (RRC) in the SAR and laser profile data. Our focus in this Section is on the relationship between the distributions of the areal coverages of the SAR RRC and HEM RRC datasets.

As is demonstrated extensively the highest backscatter values in the Baltic Sea are mostly generated by the deformed areas,

e.g., (Mäkynen and Hallikainen, 2004). The following deformation types were included: ridges, rubble fields and ice channels filled with rubble and brash ice. The ridges as well as the ice channels with rubble appear in the TSX SAR image as bright filaments. The resolution of 20 m in the TSX image is sufficient to detect partly the ridges and ice channels. For the navigational purposes the rubble filled ice channels form an obstacle for the winter shipping as discussed earlier. Hence the ice channels are not treated separately from other forms of ice deformation. The ridging as seen in the 20 m SAR data still preserves many of

the properties of the very high resolution SAR imagery. One of these properties is that the in Sect. 6.6 introduced simulation method works also for the medium resolution image.



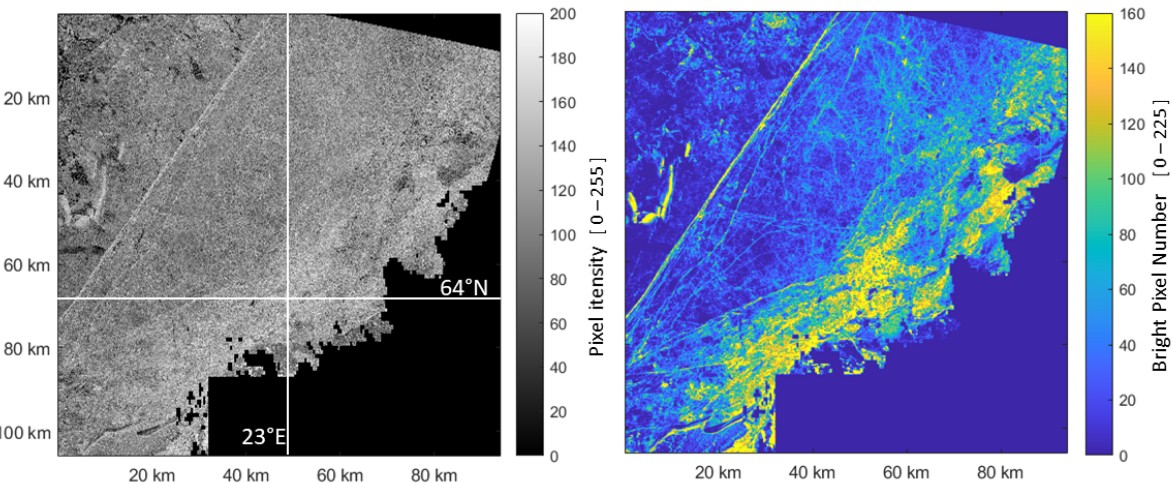

**Figure 13.** In the left the test area covered by the TSX SAR imagery with resolution of 20 m. In the right the contextual (BPN) image.

The first step in the investigation is to extract the bright pixel set with the 20% threshold (here -14.4 dB) from the SAR imagery covering the study area. This set is denoted with $B$. A binary image is constructed in which pixel has value one if it is in $B$ and zero otherwise. The resulting image is called the seed image. Similarly to the results of Sect. 5 the actual value of the threshold for $B$ is not particularly important considering our purposes; a topic we will discuss later. In the seed image the bright

pixels in the set $B$ are mostly not connected to each other. To increase the connectedness among the bright pixels and hence forming anticipated ridge structures, we convolve the seed image with a unit kernel of 15 by 15 pixels to form a contextual bright pixel number image. The amount of pixels belonging to $B$ in the convolution window is assigned to the center pixel of the window in the contextual image. This value gives the bright pixel number (BPN) when the convolution window fits inside the image while the remaining boundary area is not significant in the present context. The maximum distance to a seed image

pixel (20 m by 20 m) which contributes to the computation of the center pixel value is always less than 170 m, mostly less than 100 m. The pixel size of the resulting contextual image remains the same as in the original SAR image (20 m).

The contextual image in Fig. 13 (left panel) reveals effectively the ridged areas. The spatial arrangement of bright pixels in the seed image is such that the applied convolution operation connects many of the ridge and ice channel structures which one can detect visually in the original SAR image. The magnitude of the backscattering generated by ridging in a pixel is roughly

proportional to the size of rubble area in the pixel although $\sigma^\circ$ is subject to the random variation due to the arrangement of ice blocks (Manninen, 1992; Carlström and Ulander, 1995). Due to the curvilinear structure of a ridge and the variation of its ice block accumulations with respect to areal extent and orientation $\sigma^\circ$ fluctuates in the neighboring pixels across which the ridge meanders. The convolution operation collects these only partially visible ridge pixels surprisingly well together and is seen to act in the medium resolution SAR imagery similarly as in the high resolution imagery.





To derive ridging statistics the resolution of the contextual image is weakened to 1 km$^2$. In the coarser resolution image each pixel comprises $50 \times 50$ BPN image pixels. The larger scale image is called an analysis image. The pixel value in the analysis image is the mean of all the BPN image pixels inside the 1 km$^2$ resolution pixel and it is converted to a percentage point. These coarse pixels are the SAR RRC (SAR based ridge and rubble coverage) pixels.

The ridge rubble coverage can be estimated also from the profile data (HEM RRC). In the computation of HEM RRC ridge density is multiplied by average ridge width which is obtained from average sail height by approximation. To provide realistic estimates the effect of cutoff sail height used must be assessed as well as the applicability of the results reported on sail $width/height$ ratio which is noted $R = w/h$. Considering first the ridge width, the effect of oblique angles of sail crossings must be taken into account. If the ridges are randomly oriented, the average width is $\pi/2$ times the perpendicular width (**?**).

Hence we can assume that the ridge sail length per unit area $A = $ km$^2$ is $\pi/2$ times the ridge density. The reported $R$ ratios are usually around 4. However, these measurements typically refer to the highest point of some sail hundred or couple of hundred meters in length while aerial imagery shows that the sail width does not vary as much as the sail height but is rather constant. The results of (Lensu, 2003) indicate that the highest point of a sail is typically 2.5 times the average height of the sail. Therefore, in the present contex the value of $R$ should be increased to 10. To add, Rayleigh criterion counts as singular ridge

a certain number of wider, multiply peaked formations. Deformed ice fields usually also include scattered rubble and other diffuse roughness not accounted for in the extrapolation model. To include the contribution of these we increase the average width by further 30%. Thus the value $R = 13$ is finally used.

    The cutoff sail height affects both ridge density and ridge height. If the cutoff is raised, ridge density decreases but average sail height increases. To estimate the cutoff effect the extrapolation model by (Lensu, 2003) is applied. It is assumed that the

average sail height is representative for the whole data set and only the ridge density varies. For the present laser profile data with cutoff height 0.4 m and average sail height 0.65 m the proposed model provides estimates that the true average height is $h_a = 0.48$ m and that true densities are 2.0 times the observed density.

    Taken together, a unit increase in the observed ridge density for 0.4 m cutoff data increases the along profile rubble coverage by 1.96%:

$$\frac{RRC}{A} = \frac{\pi}{2} \times R \times h_a \times \lambda \tag{6}$$
$$= \frac{\pi}{2} \times 13 \times 0.00048 \times 2, \tag{7}$$

where all quantities are in kilometers ($h_a$=extrapolated average height, $\lambda$=ridge density extrapolation factor).

    As deformed ice fields usually include scattered rubble and other diffuse roughness not accounted for the by the extrapolation model and when we also take into account the uncertainties involved, the value of HEM RRC is rounded to 2% of the pixel

area for unit ridge density in Eq. 7. If the profiling flights have crossed an area multiple times in different directions, which was the case for coastal ridge fields near the flight base, the estimate is representative of areal rubble coverage. For a single crossing the relationship involves randomness and is generally the more reliable the larger the considered area is, provided that homogeneity of ridging conditions persists. In the statistical analysis we use the 2% estimate for HEM RRC in conversions between RRC and ridge density.



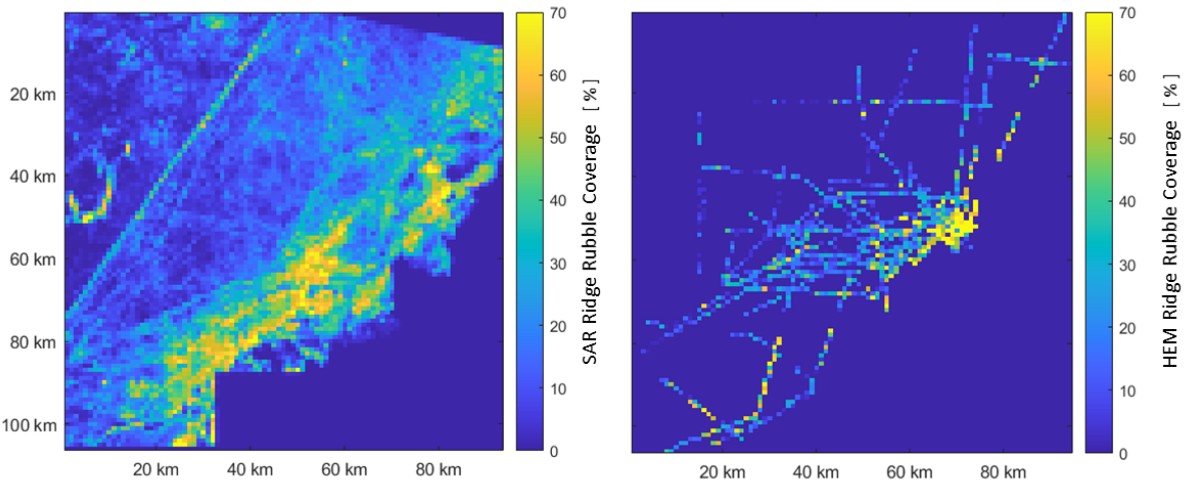

**Figure 14.** The SAR RRC image in the left panel, the HEM flights and the associated RRC values in the right panel. The unit is percentage point in both panels.

subsectionCombining SAR and Laser datasets

To each ridge the following quantities are attached: latitude and longitude coordinates as well as spacing (distance to previous ridge). From the profile data is removed the ridges with sail height less than 0.4 m. The information of the spacing is not used in the computations. A grid with 1 km$^2$ cell area for the profile measurements was formed. The coordinates of each HEM grid

cell coincidence with an analysis image pixel with the same resolution. Then the profile measurements belonging to different cells were determined. The value of a HEM grid cell is the number of ridges detected in it (RSN). HEM RRC for each grid cell is estimated as discussed above: multiplying the count of ridges by 2%. All grid cells with 50 ridges or more were estimated to have 100% RRC. About 2% of all the cells where one or more ridges occurred had 100% RRC. These cells situated near coast in the rubble field zone. It is possible that some of the ridges in these 100% RRC cells were measured more than once because

these cells were in the area where majority of the HEM flights began, see Fig. 14.

As was pointed out in Section 4.2 ice drift and additional deformation after the acquirement of the SAR image are present in the profile data but do not show up in the SAR imagery. Hence the comparison based on the one-to-one pixel correspondence is not adequate. We performed the comparison regionally between the two data sets. The sparsity of the HEM flights in many parts of the SAR image (Fig. 14) reduced the amount of the HEM RRC grid cells. All nonempty profile grid cells in the area

encompassed by the SAR image were utilised when the regional distribution of RRC based on the HEM data was computed. The SAR based analysis image is shown in Fig. 14. It covered all the profile measurement profiles and also areas between the HEM flight lines. The areal extent of the SAR image is larger than the HEM flight routes require. This is due to practical considerations. We wanted to preserve the structure of the SAR imagery. In addition ice drift made it impossible to determine the corresponding points in the SAR and HEM data with any reasonable accuracy. The ice drift of R/V Aranda varied from 0.1




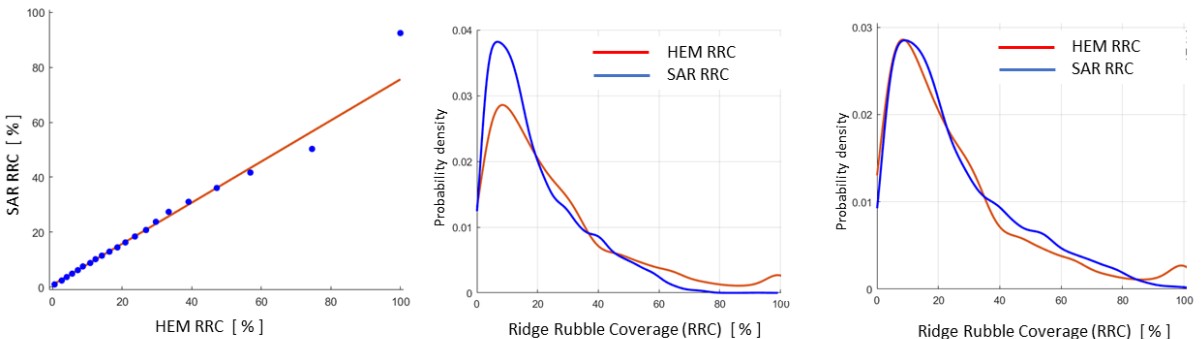

**Figure 15.** The quantile-quantile plot for the SAR and HEM RRC datasets (left), the RRC distributions (middle), the scaled RRC distributions (right).

to 0.4 knots. The time difference between SAR and HEM data ranged from two to four days. Hence the trajectory length of ice drift could be well over 10 km during the time gap. The main ice drift direction was eastward.

### 7.1.1 The results

In this Section we compare the RRC datasets based on the SAR imagery and the HEM data. The samples from both datasets
have the same resolution. We examine their similarity using the quantile-quantile (Q-Q) plot for the distributions. The use of Q–Q plots to compare two sample sets of data can be viewed as a nonparametric approach to comparing their underlying distributions.

Since the Q–Q plot compares distributions, there is no need for the values to be observed as pairs, as in a scatter plot, or that the number of samples in the two groups being compared to be of the same order of magnitude. There are about seven
times more samples from the SAR RRC estimates (N=7549) than from the HEM RRC estimates (N=1017). The distinctive and most important feature in the Q-Q plot (Fig. 15) is that the quantile pairs are largely located along a line. This implies that the distributions are linearly related which in turn implies that the shapes of the distributions are similar. Because the HEM RRC follows the gamma distribution (Eq. 3.), then also SAR RCC can be approximated by the gamma distribution. This linear relationship was not anymore valid for the upper tail of deformation estimates, approximately for SAR RRC over 50 % and
for HEM RRC over 75 % RRC (Fig. 15, the left panel). This can be expected because the SAR RRC values over 50% are relatively rare in our data. The SAR-HEM RRC relationship is also affected to some extent by the limited amount of the HEM flight lines.

From the general trend of the Q-Q plot we can see that the ratio between the HEM/SAR quantiles remained almost constant, in this case about 1.34 on the interval from 0 to 75% for the HEM data. As a consequence the distributions $f(c*x) = g(x)$
are close to each other, where $f$ is the underlying density of SAR RRC estimates, $c$ is a scaling factor (1.34) and $g$ is the



corresponding density for the HEM RRC estimates. The $1/c$ value is the slope of the line fitted to the quantiles in the HEM-SAR Q-Q plot. In the SAR-HEM Q-Q plot the slope is $c$.

We examined how close to each other the distributions of the SAR and HEM RRC are with and without the scaling factor 1.34. The estimated densities for the analysis image SAR RRC and the HEM RRC without scaling are overlaid in Fig. 15

(middle panel). The original SAR and HEM RRC densities show the same basic shape. The distributions illustrate that highly ridged areas form proportionally a much larger fraction in the HEM data than in the SAR analysis image data. This is due to the profile measurement arrangement where the HEM flights frequently flew over the large coastal rubble field zone, see Fig. 14. Consequently a significantly larger proportion of the HEM data was collected from this area than was the case with the SAR RRC data. This difference explains also the breakdown of the linear relationship in the upper tail of the Q-Q plot (Fig. 15, left

panel). One can speculate that if the HEM flights were located more uniformly over the SAR image, the scaling factor would be closer to 1.

The SAR RRC density with scaling factor 1.34 is compared to the HEM density in Fig. 15 (right panel). We observe a high agreement in the distributions. The value of the scaling factor changes if the area of the SAR image is extended. For the SAR-HEM comparison the scaling factor varied from 1.3 to 1.6 depending on how the SAR image area of interest was defined.

When the new scaling factors were applied, the resulting SAR RRC distributions were quite similar to the one shown in Fig. 15 (right panel).

The similarity of the scaled SAR RRC and original HEM RRC distributions is visually obvious. To quantify this we fit the gamma distribution to both data sets. The maximum likelihood estimates for the shape parameter of the scaled SAR RRC gamma density is 1.7 and the scale parameter 15.5. The respective values for HEM data are 1.4 and 18.2. The expected values

are very close to each other: 26.2% for scaled SAR RRC and 25.9% for HEM RRC. Converting the mean values to the amount of ridges per km$^2$ yields about 13 ridges per km$^2$ for both data sets. For the original SAR data the estimated mean is 19.5%, i.e., about 10 ridges per km$^2$. Hence the HEM/SAR ratio for the expected amount of ridges per km$^2$ without any scaling is about 1.3, i.e., the slope value in the Q-Q plot.

The essential feature found in the HEM-SAR analysis was the linear relationship between the RRC distributions from the

different sources. It was also observed that if the areal extent of SAR image with fixed BPP is increased, the scaling factor $a$ increases but the linear relationship holds. The increase of $a$ likely can be attributed to the fact that the size and the location of the HEM data remains the same in all cases but the average magnitude of SAR RRC grid cell decreases.

If we vary BPP in the identification of potential ridge pixels from the SAR imagery, the value of $c$ changes when the SAR image extent and the HEM data stay fixed. The validity of the linear relationship between these data sets is examined in three

cases: For the BPP threshold 90 % $a$ is 2.22, for 85 % 1.61 and in our analysis with the 80 % threshold $c$ is 1.34, that is, $c$ increases when BPP increases and the amount of bright pixels decreases. In all the three cases the relationship between the RRC datasets remains linear. The mean of the scaled SAR RRC distribution in these instances is always close to the HEM RRC mean. The range of the threshold values yielding largely linear relationship between these two datasets depend on the extent and concentration of HEM data. For the available datasets it is not restricted only to the BPP value used in the analysis.





## 8 Conclusions

Ice service experts are usually able to recognise the same ridging signatures in SAR images that have different resolutions or are taken in different ambient conditions. One way to to make this observation more precise is to consider binary images that select a relatively small percentage of brightest pixels from the SAR image. The ridged ice tends to appear in these as spatial

structures consisting of mostly disconnected or weakly connected pixels. Changing the percentage changes the connectivity of these pixel clouds but not their general delineation, and the same binary structures can be often identified in other SAR images with different resolutions and different acquisition parameters. We note especially the importance the acquisition frequency to the ridging identification (Eriksson et al., 2010).

The present approach proceeds from these observations by analysing SAR images in terms of local density of bright pixels

chosen by a certain percentage (BPP). The quantification of spatial variation is in terms of bright pixel numbers (BPN) counted in pixel blocks with variable side length $L$. The development considers first high resolution images for which the bright returns can be assumed to come from individual ridge sails. This provides connection to physical studies investigating the backscattering from ridge block accumulations. The approach leads first to methods to enhance the visibility of the ridging structures. The methods have a remarkable stability against changes in the BPP or in the intensity histogram. A more systematic

foundation is then provided by a hypergeometric model for BPN distribution system parameterised by $L$ and BPP. The model is derived by considering the rate of change for BPN in a generative process where the BPP is increased in small steps. If the rate of change for increases linearly with the BPN, the model ensues. The same approach applies to ridge sail numbers (RSN) in profile data segments of length $L$, or in essential to the description of spatial variation of ridging in terms of local ridge density. Here the generative process decreases in small steps the cutoff sail height above which the sails contribute to the RSN

values.

The hypergeometric distribution model was validated both for RSN and BPN statistics by first validating the linearity assumption of the generative process and then by the goodness of distribution fit. The BPN distribution model was further corroborated by a image texture simulation that followed the generative assumption. The results put the SAR image statistics and the ground truth ridge density statistics on a unified base, reducing the conversion problem to the establishing of parameter

relations between the models. The statistical connection was then studied for the medium (20 m) resolution image for which concurrent ground truth RSN data was available. The same gamma distribution, apart from a scaling constant, was found for both.

The results raise the question on the relation of the observed statistics to the actual deformation process. Ridging process in a converging ice field manifests along a surface profile as physical appearing of new ridge sails to profile segments. This is

analogical to the generative RSN process where the ridges appear to the segments due to decrease of cutoff sail height. Thus the hypothesis will be that the ridging as a sequential process of ridge formation events is behind the observed statistics. In this context a more common approach to spatial sail statistics in terms of sail spacings or sail-to-sail distances needs be considered. The formation of a new sail cuts the spacing into two, and the process is analogical to a sequential fragmentation process. If the probability for this to happen does not depend on the spacing, the spacing distribution is asymptotically lognormal as





originally shown by (Kolmogorov, 1941). The lognormal has also been found to apply to sail spacings, in the Baltic since (Lewis et al., 1993). Conceptually, neglecting the strains, lognormal generative hypothesis entails the generative hypothesis of the hypergeometric model for larger RSN.

It is worth noting that if the preceding is accepted with its face value the ridge formation rate increases with ridge density and consequently the thickness increases with thickness. This runs counter the usual assumptions of thickness redistribution. However, the appearing of a sail to a 1D profile segment is a manifestation of a 2D ridge formation event that may extend even kilometres in along sail direction. A possible and more plausible 2D explanation is that the initial fault triggering the ridge buildup is more likely to find its way into already damaged areas with material discontinuities. However, these questions are almost uncharted territory. The fact that the scale of ridge formation events typically exceed the pixel block scales of the present study may also explain that the statistical model for BPN was parametrizable by $L$ rather than $L^2$. The power law decrease of parameter $a$ with $L$ has then exponent that is about the same for both 2D BPN and 1D RSN data. The power law itself suggests that the distributions have fractal support. If the Poisson deposition rate $a$ is effective only within support, its value decreases with the increasing scale $L$ of the block coverage approximating the support. Also in the simulation procedure in Section 6.6, it was found that better results were obtained by reducing the effect of $a$ in likely level ice areas.

Ideal data for the present approach would consist of SAR data with different resolutions, acquisition parameters and ambient conditions over same ridged ice areas, and matching ice surface topography data from laser scanning, visual wavelength images and other methods. In the present study the ground truth data was available only for the medium resolution SAR image with some temporal separation and approximate spatial matching, and the connection between the high resolution image and ground truth remained statistical, although firm. This leaves several lines of study unpursued. Perhaps the most important is the relationship of ridging signature in SAR images with a different resolutions. In the high-resolution (1.25 m) image the bright returns can be related to the presence of ridge rubble and to what is known about backscattering from block accumulations. For discussion purposes, a bright pixel in the medium resolution (20 m) image can be thought of as a $L_j = 16$ pixel block of high-resolution (1.25 m) pixels. A certain number $n_j$ of pixels in the block are bright returns from ridge rubble and the remaining are non-bright returns both from ridge rubble and other surface types. The distribution $k(n_j)$ of high resolution bright returns is certain to have pertinence to the ridging signatures of the medium resolution image. Thus the hypergeometric scale system can also provide tools for linking different SAR resolutions and for anchoring the observed intensities to the physical backscattering returns from ridge rubble.

*Data availability.* The laser data is available at PANGAEA under the following DOI: https://doi.pangaea.de/10.1594/PANGAEA.930545 (Haas et al., 2021).

*Author contributions.* ML has main responsibility on analysis and writing in Sections 5 and 6, and MS in Section 7. The remaining sections are joint work.





*Competing interests.* The authors declare that they have no conflict of interest.

*Acknowledgements.* This work was supported by EU, Finland, Norway, the Russian Federation, and Sweden, through the Kolarctic Cross Border Collaboration Project "Ice Operations" under Grant KO2100 ICEOP.



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
