# Peer review of "Ice ridge density signatures in high resolution SAR images"

_The Cryosphere, 2021_

## Author Response (AR1)

**Comments to the author:**

Dear authors,

Thank you for your comments to the reviewers.

The two reviewers note that the paper presents a new method to derive sea ice ridge height using TerraSAR-X imagery, but they also point out some important questions. Particularly, the text should be shortened, the structure of the paper should be reorganized, the impact of warm air temperature should be better explained, and how the work can be used to aid safe navigation should be better explained. These are all important comments to improve the quality of the paper. I strongly encourage you to take into account these comments in the revised paper.

Best,

Dr. Kang Yang Editor, The Cryosphere

**Response by the authors**

Dear Dr. Yang

Thank you for you comments and encouragement.

Following your suggestions and also those of the Reviewers our paper has been thoroughly revised. It has been shortened so that the length is about 65% from the previous length. Data/methods, results and discussion/conclusions are now in their own sections (2,3 and 4) and we have rearranged and edited all material accordingly, removing all repetitions. The Introduction has been shortened to a length that is typical in TC papers, and the same applies to Dicussion and Conclusions. As suggested by the Reviewers, several groups of short sections have been combined into longer ones and the text flow has been improved by combining short paragraphs and by combining short sentences into longer ones. The text has been also gone trough a language check (British english). We have also gone trough the minor comments of the Reviewers once more when revising our paper.

We also included a separate section on the use of the results in ice information production, and addressed the effect of above zero temperatures and wet snow in the Introduction and Discussion. However, we remark also that this issue cannot be conclusively settled otherwise than by applying the approach to SAR image pairs over the same ice fields and taken before and after the change in ambient conditions.

In addition, the graphics has been in part redrawn We think that the present Figures add essentially to the accessibility of the paper and hope that they can be incorporated as the paper is otherwise of reasonable length. Due to the comprehensive changes the difference file is not very informative, but we have uploaded it anyhow.

Best Regards,

Mikko Lensu and Markku Similä

---

## Author Response (AR2)

**Responses to Reviewer #1**

At the beginning of section 1, the author addressed a need for ridge parameter retrieval from SAR imagery. However, a clear statement on the research objective is missing. Research objectives and paper outlines should be clearly mentioned at the end of section 1 so that a reader can be informed on what to expect in the later parts of the manuscript.

*We have added a short standard summary of the aims and paper disposition to the end of introduction.*

In section 2.1, it is mentioned that wind speed was 18 m/s during the campaign, which can substantially alter the snow distribution compared to the SAR acquisition date – a week prior. This is further evident from the higher std dv of snow thickness. My concern is how well the airborne data (acquired after a week) represent snow thickness during SAR acquisition. What are the uncertainties, and how does that affect the results? A recent paper in TC Discussion investigated wind re-distributed snow effect on backscatter from the MOSAiC expedition, and the effect was found substantial.

*These questions cannot be answered by short additional comments in the paper only. As the topic of snow is highly relevant both SAR backscattering and to the interpretation of profiling data we provide a more extensive discussion response.*

*The 18 m/s is the maximum persistent (1h average) wind observed during the campaign and there were several periods, most significantly on 24-25.2, 2.3 and 3.3. The wind speeds are from RV Aranda and consistent with coastal weather station data. For snow transport considerations relevant data was obtained also from a field weather station operating from 27.2 to 10.3. The sensor was about 1 m above the surface and about 17 m/s maximum 1 h average wind speeds are observed for the days 2.3 and 3.3. The snow data (1500 m of calibration line, 100 m ridge crossing and a 20x20 m grid) were measured during the field camp period from 27.2 to 3.3 and any significant aeolian snow transport would have been observed if it would have been triggered by the stronger winds.*

*In the mainland the snow conditions were very uniform between the latitudes corresponding to the HEM campaign and thus the field station snow data can be assumed to be representative to the whole HEM campaign as well. The mainland snow thickness increased from 50 cm on 1.2 to 60 cm on 6.2. After that there was no significant snowfall before 11.3. Periods of strong winds occurred in February also prior the campaign. The temperatures were below zero in February so that during the first week of February and earlier the snow layer on sea ice was dry and easily carried by the wind. In aerial images taken during the campaign snow dunes are visible, accumulated behind ridge sails and typically extending from them in oblique angles while in the ridge sails the block structure is mostly visible. The dunes crossed by snow lines were typically at most 40 cm high and had roughly triangular cross sectional shape. The density was typically 400 and up to 450 kg/m3. Although morphologically similar the dunes were not yet sastrugi but consisted of densely packed snow sculpted by wind and resistant against wind erosion. The aerial images also showed that extended level ice areas could be in part snow free. Thus the conclusion is that, due to the preceding long period with no snowfall, the aeolian snow transport was not*

*significant any more during the campaings and the snow cover morphology persisted more or less unchanged.*

*As concerns the effect of snow in HEM data, the determination of the reference level for the ridge height is based on standard procedure where minimum profile values are identified from profile segments and accepted as zero elevations. In closed ice cover of the campaign these were likely to be located in level ice areas and have snow thickness typically from 0 to 10 cm. So the reference level has typically freeboard from 5 to 15 cm although is assigned zero value. This must be considered in rubble mass balance as the ridge sails are higher than they appear in the data. This affects also the estimation of sail rubble coverage from the sail data but in our estimates the correction can be assumed to be included to the generous rounding upwards. In statistical considerations the presence of reference level freeboard is equivalent to a reduction of the ridge height cutoff value. Thus the applied cutoff 0.4 m is likely to be close to 0.5 m cutoff in reference to the water surface. The cutoff also leaves out most of the snow dunes as these extended typically at most 0.4 m above water level. The presence of the dunes was also visible in the statistics below the cutoff and aerial images also indicated that the density of snow dunes can be locally five to ten times the density of ridges at the same location.*

Minor comments

In-text reference needs to be checked for proper formatting. For example, Page 2, line 14 should be Manninen (1992 and 1996). Same comments for page 2, line 18 and 20. Check for similar issues throughout the manuscript.

*Both referees pointed to our careless use of the references. Now they are checked and corrected. Brackets around the year, e.g., Manninen (1992), were used if the reference is the subject in the sentence or if a preposition precedes the reference. Otherwise we included also the surnames inside the brackets, e.g., (Manninen, 1992).*

Fig. 1: I find the text in the figure very small and difficult to read. Three additional southern stations are shown, which falls outside of the airborne/SAR coverage, thus can be removed. I also suggest using the same scale and extent for all three sub-figures. Put a scale on the maps.

*The ice charts are now in the same scale as the flight track map, and scale has been added. We also completed the figure caption as requested by Reviewer #2.*

Fig. 5: Denote the blue and yellow box in caption.

*Corrected.*

Fig. 6: What is the white box at top-left corner of the image? What are the numbers in the legend of BPN percentile class? Are these number of pixels within each category? Update the caption accordingly.

*We amended:*

The 15200x15200 contextual image (on the left) and corresponding category image derived from the full image with BPP 20% and *sliding block side length L=101 pixels. The colorbar extends to the maximum observed BPB in the blocks, 9831. The location of the 1024x1024 subimage is also indicated*

Page 18, line 5-9
"…. Does not change much". Several qualitative statements are made in this section. I suggest putting a quantitative measure (perhaps in %) to provide a convincing message to the reader. Also, run a significant test to state that the changes are not-/significant.

*These statements were intended to be a preamble to the subsequent text demonstrating the insensitivity of contextual images against BPP changes; this section treats the matter quantitatively also. The text was rephrased to avoid misunderstandings.*

**Responses to Reviewer #2**

The manuscript is in my opinion significantly improved compared to the first submission. Some minor changes should still be implemented but after that I would recommend publication.

P2 R1-3. Perhaps include some example references here.

*This section is better suited to precede the review of Baltic research somewhat below that also includes appropriate references. So it was moved there, rephrasing* 'In the Baltic, SAR research has approached…'

P2R6. Please be specific if you are using Celsius, Kelvin or Fahrenheit when referring to temperatures.

*Done in this and few other locations.*

P2R21. Should (Similä… be included in the sentence before? If not, then the sentence shouldn't start with a bracket. Overall have a good read through to see where the references should include a bracket around the year and where it should be also including the surnames. As it is a great many sentences starts with a bracket.

*The references have been corrected, see response to Reviewer #1 on the same matter.*

P3R23. .3 -> 0.3  *Corrected*

Fig 1. This figure includes illustrations of more than just the flight lines from 2011. Please update this figure to include a caption outlining all three subfigures.

*We completed the caption. See also the response to Reviewer #1 on the same figure.*